



# Biophysical effects on the interannual variation in carbon dioxide exchange of an alpine meadow on the Tibetan Plateau

**Lei Wang[1], Huizhi Liu[1], Jihua Sun[2], and Yaping Shao[3]**

1 LAPC, Institute of Atmospheric Physics, Chinese Academy of Sciences, Beijing 100029, China

2 Meteorological Observatory of Yunnan Province, Kunming 650034, China

3 Institute for Geophysics and Meteorology, University of Cologne, Cologne, 50937, Germany

*Correspondence to:* Huizhi Liu (huizhil@mail.iap.ac.cn)

**Abstract.** Eddy covariance measurements from 2012 to 2015 were used to investigate the interannual variation in carbon dioxide exchange and its control over an alpine meadow on the southeast margin of the Tibetan Plateau. The annual net ecosystem exchange (NEE) from 2012 to 2015 was -114.2, -158.5, -159.9 and -212.6 g C m$^{-2}$ yr$^{-1}$ and generally decreased with the mean annual air temperature (MAT). An exception occurred in 2014, which had the highest MAT. This was attributed to higher ecosystem respiration (RE) and similar gross primary production (GPP) in 2014 because the GPP increased with MAT but became saturated due to the photosynthesis capacity limit. In the spring (March to May) of 2012, lower air temperature ($T_a$) and drought events delayed grass germination and reduced GPP. In the late wet season (September to October) of 2012 and 2013, the lower $T_a$ in September and its negative effects on vegetation growth caused earlier grass senescence and significantly lower GPP. This indicates that the seasonal pattern of $T_a$ greatly affected the annual total GPP, which is consistent with the result of the homogeneity-of-slopes model. The model shows that the climatic seasonal variation explained 48.6% of the GPP variability, and the percentage of climatic interannual variation and the ecosystem functional change were 9.7% and 10.6%, respectively.

**Keyword:** Carbon dioxide exchange; Interannual variation; Alpine meadow; Tibetan Plateau





## 1 Introduction

In the last decade, the carbon dioxide exchange in grassland ecosystems has attracted much attention (Aires et al., 2008; Baldocchi, 2008; Hunt et al., 2004; Suyker et al., 2003) because grasslands cover 32% of the global land surfaces and may contribute greatly to the carbon cycle on a global scale (Parton et al., 1995).

The annual total net ecosystem exchange (NEE) for grasslands has a large range from -650 g C m$^{-2}$ year$^{-1}$ to 160 g C m$^{-2}$ year$^{-1}$ due to climate variability and land use changes (Gilmanov et al., 2007; Wang et al., 2016a). The climatic factors of $CO_2$ exchange also vary under various climate conditions (Du and Liu, 2013; Fu et al., 2009; Xu and Baldocchi, 2004). Most of previous studies focused on the low-lying grasslands (Gilmanov et al., 2010).

The alpine meadow in China is the primary grassland type of the nation and is mainly distributed in the Qinghai-Tibetan plateau (DAHV and CISNR, 1996; Liu et al., 2008). The warming trend in high-altitude areas has been observed to be more pronounced, such as the Tibetan Plateau and its southeast margin (Fan et al., 2011; Liu and Chen, 2000). Several studies of $CO_2$ exchange on the Qinghai-Tibetan plateau have been performed, where the mean annual $T_a$ is approximately $0^{\circ}C$ (Gu et al., 2003; Kato et al., 2006; Shi et al., 2006;

Zhao et al., 2006). The daily $CO_2$ fluxes of the alpine meadow-steppe in Damxung, Tibet were jointly affected by air temperature and soil moisture (Fu et al., 2009), while the daily $CO_2$ fluxes of an alpine shrubland at Haibei, Qinghai were sensitive to air temperature (Zhao et al., 2006). On an annual scale, the measurements at the Haibei alpine meadow revealed that the annual $CO_2$ uptake was increased by the earlier onset of the growing season, which was caused by higher air temperature (Kato et al., 2006). The Lijiang

alpine meadow is in a much warmer area (the mean annual $T_a$ is $12.7^{\circ}C$). A spring drought event and relatively low soil moisture significantly delayed the start time of grass germination and reduced the annual $CO_2$ uptake (Wang et al., 2016b). How the annual $CO_2$ exchange responds to the mean annual air temperature (global warming) has not been clear for the alpine meadow ecosystems.

As previous studies proposed, year-to-year changes in $CO_2$ exchange are attributed to climatic variability

(Hui et al., 2003; Xu and Baldocchi, 2004). Fluxes may directly respond to climatic drivers or be indirectly affected by the functional changes or the changes in the flux-climate relationships (Polley et al., 2008). Statistic models have been used to partition the interannual variation (IAV) of the $CO_2$ exchange (Hui et al., 2003; Richardson et al., 2007; Teklemariam et al., 2010). For example, Shao et al. (2014) found that 77% of the observed variation in NEE was explained by functional changes in the moist grassland in USA, while

variations of climatic variables could better explain the IAV of NEE for a meadow in Denmark and (Jensen et al., 2017) and mixed-grass prairies in the semiarid area of USA (Polley et al., 2008). The relative importance of the direct and indirect effects of the climatic variables on the interannual variations in $CO_2$ exchange for the alpine meadow in China has not been quantified.

The $CO_2$ exchange between the atmosphere and the Lijiang alpine meadow was measured using eddy



covariance technique from 2012 to 2015. The objectives of this study include: (1) to examine the seasonal

and interannual variation in NEE, GPP, RE and the parameters of ecosystem photosynthesis and respiration;

(2) to investigate the main environmental controls of the total GPP, RE and NEE on the seasonal and annual

scales; and (3) to partition the interannual variation in GPP, RE and NEE into climatic variability and

vegetation growth.

## 2   Observation site and methods

### 2.1   Observation site

The observation site (27°10'N, 100°14'E, 3560 m a.s.l.) is located at Maoniuping of Yulong Snow

Mountain to the north of Lijiang City on the Tibetan Plateau, China. The study area is under the plateau

monsoon climate, which is influenced by the southwest and southeast monsoons. The wet and dry seasons are

clear, and the wet season is from June to October. The 30-year mean annual total precipitation (1981-2010) at

Lijiang City (2400 m a.s.l.) is 980.3 mm, and 85% of the precipitation is concentrated in the wet season. The

30-year MAT is 12.6°C (data from the Lijiang Meteorology Bureau). The dominate species of this alpine

meadow are *Kobresia Willd* grass, with a maximum height of 20 cm, and *Berberis Linn* shrub, with a

maximum height of more than 60 cm. The surface is covered by green vegetation, litter and bare soil. The

soil type is loamy soil with a dark brown color, which has lower reflectance than the grass canopy (Guo et al.,

2009).

### 2.2   Field measurements and normalized difference vegetation index (NDVI)

The eddy covariance (EC) system was used to measure 3-D wind speed and the $H_2O$ and $CO_2$

concentrations at a height of 2.5 m with 10 Hz frequency. The system consists of a three-dimensional sonic

anemometer (CSAT3, Campbell, USA) and an open-path $CO_2/H_2O$ infrared gas analyzer (LI-7500A, LI-COR,

USA). The low response measurements (1/3 Hz frequency) performed in this study include air temperature

and relative humidity at a height of 2.5 m close to the EC system (HMP45C, Campbell, USA). Net radiation

(including shortwave and longwave radiation, CNR4, Kipp&Zonen, Netherlands) and photosynthetic active

radiation (LI190SB, LI-COR, USA) were measured at 1.5 m. Soil temperature (109-L, Campbell, USA) and

soil water content (CS616, Campbell, USA) were measured at a depth of 5 cm below the ground. The

precipitation (including solid precipitation in the winter) was measured using a weighing bucket precipitation

gauge (T-200B, Geonor, Norway). All measurements were controlled by a data logger (CR3000, Campbell

Scientific, USA), and the data were stored on a 2-GB CF card.

Four points around the flux tower were selected to investigate the variations in vegetation growth. The

$250 \times 250$ m$^2$ gridded NDVI data at 16-day intervals (product name: MOD13Q1) for the four points were

obtained from the Moderate Resolution Imaging Spectrometer (MODIS) on the EOS-1Terra satellite and





were averaged to represent the meadow in this observation site.

### 2.3 Flux calculation and quality control

EddyPro software (version 5.1, LI-COR, USA) was used to calculate the half-hourly $CO_2$ flux based on the 10 Hz raw data. After a spike detection (Vickers and Mahrt, 1997), the sector-wise planar fit method was used to transform the coordinate system due to a terrain slope of approximately 10° (Wilczak et al., 2001).

Other corrections for $CO_2$ flux include spectral loss correction (Moore 1996,) and density correction (WPL correction) (Webb et al., 1980).

Stationary and integral turbulence characteristics tests were used for flux quality control (Foken and Wichura, 1996). When u* was less than 0.1 m s$^{-1}$, the $CO_2$ flux was dependent on u* and was discarded. Since there is a coniferous forest approximately 350 m to the north of the site, an analytical footprint model

was used to determine whether the half-hourly $CO_2$ flux is influenced by the forest and needs to be removed (Kormann and Meixner, 2001).

After quality control, approximately 70% of the $CO_2$ fluxes were subjected to further analysis. Linear interpolation was used to fill the flux gaps of less than 2 hours. To fill gaps longer than 2 hours, marginal distribution sampling, an improved 'look up table' method, was used (Falge et al., 2001; Lloyd and Taylor,

1994)

### 2.4 Data analysis

The relationship between daytime NEE ($NEE_{daytime}$) and PAR was described by the Michaelis-Menten model (Falge et al., 2001):

$$NEE_{daytime} = \frac{\alpha NEE_{sat} PAR}{\alpha PAR + NEE_{sat}} + RE_{bulk} \qquad (1)$$

where $NEE_{sat}$ is the NEE at the saturated light level, $\alpha$ is the apparent quantum yield (μmol $CO_2$ μmol$^{-1}$ photons) and $RE_{bulk}$ is the bulk estimated RE.

The Van't Hoff equation was used to evaluate the relationship between the nighttime NEE ($NEE_{nighttime}$, μmol $CO_2$ m$^{-2}$ s$^{-1}$) and soil temperature at a depth of 5 cm ($T_s$, °C) (Aires et al., 2008):

$$NEE_{nighttime} = a \exp(bT_s) \qquad (2)$$

where a and b are the regression parameters. The temperature sensitivity coefficient ($Q_{10}$) of RE was determined using the following equation.

$$Q_{10} = \exp(10b) \qquad (3)$$

The partitioning of NEE into GPP and RE was based on the assumption that the sensitivity of RE to soil

temperature was the same during the day and at night (Falge et al., 2001). The regression parameters derived





from the nighttime data were extrapolated to the daytime to calculate the daytime RE and the daily RE. The daily GPP was calculated as follows.

$$GPP = RE - NEE \qquad (4)$$

## 3   Results

### 3.1   Weather conditions and NDVI

The daily integrated solar radiation ($S_{in}$) varied from 1.15 to 32.40 MJ m$^{-2}$ d$^{-1}$ (Figure 1a). The mean $S_{in}$ in spring (March to May) was 17.0 to 19.93 MJ m$^{-2}$ d$^{-1}$ and was obviously larger than those in other season. In the wet season, the mean $S_{in}$ was 9.99 to 11.05 MJ m$^{-2}$ d$^{-1}$.

The mean annual air temperature ($T_a$) was 5.92 to 6.32$^{\circ}$C (Table 1). The daily mean $T_a$ ranged from 0.41 to 14.96$^{\circ}$C in the wet season and decreased to the minimum value of -9.06$^{\circ}$C in the winter. In contrast, the soil temperature never decreased below 0$^{\circ}$C, and the maximum value was 16.48$^{\circ}$C (Figure 1b). The vapor pressure deficit (VPD) reached its maximum value of 1.07 kPa before the wet season (Figure 1c). The VPD decreased to near 0 kPa, and the mean VPD for the wet season was 0.125 to 0.166 kPa.

The annual precipitation from 2012 to 2015 ranged from 1066.1 to 1257.4 mm. The precipitation for the wet season ranged from 906.1 to 1092.6 mm, accounting for 85% to 91% of the annual total precipitation (Table 1). The mean annual soil water content (SWC) had small interannual variability, from 0.227 to 0.233 m$^3$ m$^{-3}$. In the wet season, SWC reached its maximum value of approximately 0.35 m$^3$ m$^{-3}$, and the minimum SWC was 0.15 m$^3$ m$^{-3}$ (Figure 1d).

The NDVI for this alpine meadow showed clear seasonal and interannual variation (Figure 1e). The NDVI exceeded 0.4 at the end of April or late May, depending on the amount and distribution of precipitation in the spring (March to May). The maximum NDVI for each year was 0.72 (2013) to 0.60 (2012). In all four years, NDVI decreased below 0.4 at the end of October.

### 3.2   Seasonal and interannual variations in NEE$_{sat}$, α and Q$_{10}$

The daytime NEE and PAR were averaged with the PAR bins of 100 μmol m$^{-2}$ s$^{-1}$ to avoid random errors. For each month in the wet season, the daytime NEE decreased with PAR until reaching a critical PAR. Above the critical PAR, the daytime NEE increased and the $CO_2$ uptake was depressed (Figure 2a). To derive NEE$_{sat}$ and α, the NEE and PAR data were used only when PAR was below the critical value. NEE$_{sat}$   showed clear

seasonal variation (Table 2). The mean NEE$_{sat}$ values for each month show that NEE$_{sat}$ increased starting in June (-11.59 μmol m$^{-2}$ s$^{-1}$) and reached its maximum value in August (-20.14 μmol m$^{-2}$ s$^{-1}$). The highest NEE$_{sat}$ during the whole observation period occurred in August of 2014 (-23.75μmol m$^{-2}$ s$^{-1}$). NEE$_{sat}$ then declined with grass senescence in September and October. The NEE$_{sat}$ in October   (-9.36 μmol m$^{-2}$ s$^{-1}$) was





less than half that in August. The interannual variations in $NEE_{sat}$ was also large. For example, $NEE_{sat}$ in
September 2015 (-21.44 µmol $m^{-2}$ $s^{-1}$) was almost twice that in September 2013 (-11.43 µmol $m^{-2}$ $s^{-1}$; Table 2).
On the monthly scale, 81% of the variation in $NEE_{sat}$ can be explained by the mean NDVI (Figure 2b). Over
this meadow, $NEE_{sat}$ did not correlate with SWC significantly because the soil water conditions were always
good in the wet season.

At monthly intervals, there were large random errors in the regression between RE and $T_{soil}$. For example,
the $R^2$ for each month of the wet season in 2012 was 0.04 to 0.12. Thus, in 2012, the data in the wet and dry
season were combined to fit the regression (Figure 3a). The $Q_{10}$ in the wet seasons was similar,
approximately 3.45 (Table 3), which was in the normal range of previous studies (1.2 to 3.7; Falge et al.,
2001). These values were obviously higher than those for temperate grasslands (1.99 to 3.07; Wang et al.,
2016a), Mediterranean grasslands (1.22 to 2.36; Airest et al., 2008) and the Haibei alpine meadow (1.50 to
2.27; Kato et al., 2004). $Q_{10}$ was obviously lower in the dry season than in the wet season.

### 3.3 Seasonal and interannual variation in NEE, GPP and RE

The ecosystem started to absorb $CO_2$ (negative value of NEE) on DOY 165 in 2012, DOY 137 in 2013,
DOY 116 in 2014, and DOY 104 in 2015; then, NEE decreased (Figure 4). The minimum daily NEE for each
year occurred in July or August (-3.52 g C $m^{-2}$ $d^{-1}$ on DOY 196 in 2012, -3.35 g C $m^{-2}$ $d^{-1}$ on DOY 218 in
2013, -3.43 g C $m^{-2}$ $d^{-1}$ on DOY 243 in 2014, and -4.16 g C $m^{-2}$ $d^{-1}$ on DOY 210 in 2015). NEE increased
significantly starting in September and became positive on DOY 293 in 2012, DOY 305 in 2013, DOY 295 in
2014 and DOY 297 in 2015. The maximum difference in the start time of $CO_2$ uptake was 61 days while the
difference in the end time was 12 days. The $CO_2$ uptake period was much shorter in 2012 (129 days) than in
2013 (169 days), 2014 (180days) and 2015 (194 days).

The daily GPP increase started earlier than $CO_2$ uptake. The seasonal pattern in daily GPP was similar to
that of NEE, and the amplitude of GPP was larger than that of NEE. The maximum daily GPP for each year
was 6.02, 5.47, 6.23 and 5.95 g C $m^{-2}$ $d^{-1}$ from 2012 to 2015. Compared with the NEE and GPP, the seasonal
variation in RE was smaller during the wet season. In particular, RE varied slightly from June to August.

The annual GPP in 2014 and 2015 was obviously higher than in 2012 and 2013 due to the larger NDVI
(Figure 5; Table 1, 4). In contrast, RE was highest in 2014 among the four years because it had similar $Q_{10}$
but the highest air temperature (Table 1). Therefore, the annual NEE in 2014 was similar to that in 2013 but
lower than that in 2015, although the GPP was similar in 2014 and 2015. The spring drought produced a
significantly lower NDVI in 2012; consequently, the annual GPP in 2012 was the lowest. The annual NEE
for the four years followed the order of 2015<2014<2013<2012 (Table 4), consistent with the length of the
$CO_2$ uptake period.





## 4 Discussion

### 4.1 Partitioning the interannual variation in $CO_2$ exchange

The homogeneity-of-slopes model was used to partition the interannual variation (IAV) in $CO_2$ exchange
into climatic variability and ecosystem functional change, which is reflected by the variability of the
flux-climate relationship among years (Hui et al., 2003). During the wet season, the daily NEE, GPP and RE
were mainly related with $T_a$ (Figure 6). The effect of PAR on NEE and GPP was very weak, with R values of
-0.05 and 0.08, respectively.

A separate-slopes model was constructed for each year, and the multiple regression model was based on
data from the observational period. Compared with the multiple regression model, the separate-slopes model
improved the NEE estimation substantially, with $R^2$ increasing from 0.69 in the multiple regression model to
0.79 in the separate-slopes model. This means that the separate-slopes model accounted for 10.3% more
variation in the observed NEE than the multiple regression model, which is attributed to the functional

change ($SS_f$). The other 89.7% of the variation in the observed NEE was partitioned to interannual climatic
variability ($SS_i$, 7.7%), seasonal climatic variation ($SS_s$, 37.7%), and random error ($SS_e$, 44.3%) (Table 5).
Therefore, most of IAV in NEE, GPP and RE was attributable to variation in climatic variables, in particular,
climatic seasonal variation. This is in line with the findings for a *Skjern* meadow in Denmark and a temperate
ombrotrophic bog in Canada (Jensen et al., 2017; Teklemariam et al., 2010). In contrast, Braswell et al. (1997)

and Shao et al. (2014) found that functional change accounted for more IAV in fluxes than did direct effects
of IAV in climate. Moreover, the contributions of different drivers to the IAV in GPP was similar to that of
NEE, while the functional change in RE was twice that of NEE and GPP. The $R^2$ for NEE, GPP and RE in the
multiple regression model was 0.44, 0.53 and 0.59, respectively. It was reasonable that the random error of
NEE was the largest.

### 4.2 Control of the interannual variation in the $CO_2$ exchange

To examine the interannual variation in $CO_2$ exchange, the cumulative NEE, GPP and RE in 2013, 2014
and 2015 were compared with those in 2012 (Figure 7). The cumulative $NEE_{diff}$ (the difference in NEE)
values for 2014-2012 and 2015-2012 increased rapidly in spring and autumn. In summer, the differences

among 2012, 2014 and 2015 varied slightly. Starting in April, the cumulative $NEE_{diff}$ for 2013-2012 increased
until early August. These patterns were similar to those for $GPP_{diff}$. However, the annual cumulative $GPP_{diff}$
(24.3 to 147.2 g C $m^{-2}$ $yr^{-1}$) was relatively larger than the annual cumulative $NEE_{diff}$ (-44.2 to -98.3 g C $m^{-2}$
$yr^{-1}$). The cumulative $RE_{diff}$ decreased from DOY 1 and then increased in spring. The cumulative $RE_{diff}$ for
2013-2012 and 2015-2012 reached its maximum at the end of June, while the cumulative $RE_{diff}$ for

2014-2012 increased throughout the entire year, and was the largest.

The daily $CO_2$ uptake over this meadow ecosystem increases with $T_a$ (Wang et al., 2016). Especially in the





spring (March to May), the temperature environment affected the vegetation growth and GPP. From March to May, the cumulative $T_a$ was 592.3, 577.1, 633.1 and 647.6$°$C from 2012 to 2015. Consequently, the cumulative GPP in the spring increased in the order 2015, 2014 and 2013. The exception was that the spring

of 2012 had higher $T_a$ but lower GPP than the spring of 2013. Compared with the GPP in 2013, the GPP in 2012 increased more significantly due to the higher $T_a$ from March to April. However, the drought in May 2012 delayed vegetation growth and reduced GPP. The difference in $GPP_{cum}$ in 2013-2012, 2014-2012 and 2015-2012 at the end of May was 20.0, 63.1 and 83.3 g C m$^{-2}$, representing 82.6%, 42.9% and 59.7% of the difference for the entire year.

From July to October, the NEE, GPP and RE were all strongly correlated with $T_a$ on a monthly scale ($R^2$=0.84, 0.86 and 0.73, respectively) (Figure 9a, b, c). The slope between GPP and $T_a$ was much larger than that between RE and $T_a$, indicating that when $T_a$ increased, the alpine meadow ecosystem absorbed more $CO_2$. The monthly GPP in July and August varied slightly among the four years, while the interannual variability of the GPP in September was the largest because the monthly mean $T_a$ in September for 2012 (8.6$°$C) and

2013(8.8$°$C) was significantly lower than those for 2014(9.7$°$C) and 2015(10.0$°$C). Consequently, the difference in $GPP_{cum}$ in 2014-2012 and 2015-2012 from September to October was 55.2 and 48.2 g C m$^{-2}$, representing 37.5% and 34.6% of the difference for the entire year.

On the annual scale, the annual total NEE decreased (2012, 2013 and 2015) with mean annual $T_a$ (MAT) and then increased when MAT was the highest in 2014. The reason for this was that the annual total RE

increased linearly with MAT ($R^2$=0.97), while the relationship between the GPP and MAT was non-linear (Figure 9d). The GPP became saturated with increasing MAT. In contrast, the annual NEE increased with MAT at the Haibei alpine meadow, although the annual NEE was comprehensively controlled by the temperature environment (Kato et al., 2006).

**4.3  Comparison of annual CO₂ exchange with other sites**

In addition to temperature effects, the daily RE was also correlated with the daily GPP (RE=0.44GPP+0.63, $R^2$=0.82) during the wet season from 2012 to 2015. The percentage of RE to GPP for this meadow site was lower than those of Mediterranean grasslands (RE=0.53GPP+0.72, $R^2$=0.85, Aires et al., 2008; RE=0.47GPP+1.33, $R^2$=0.85, Xu and Baldocchi, 2004).

On the annual scale, the GPP at the study site were much larger than those for semiarid grasslands in Tibet and Canada (Flanagan et al., 2002; Yu et al., 2006), but much lower than the moist grasslands in low-lying areas (Table 7). Due to the high elevation and low soil temperature in the summer, the lower RE caused similar or even lower annual NEE (mean value: -161 g C m$^{-2}$ yr$^{-1}$) at Lijiang than the moist grasslands with low elevation (Table 7). For example, the mean annual NEE for the meadow in Denmark (annual

precipitation: 809 mm) was -156 g C m$^{-2}$ yr$^{-1}$, while the mean annual NEE for the C3/C4 grassland in Japan





(annual precipitation: 1156 mm) was -17 g C m$^{-2}$ yr$^{-1}$. The ratio of RE to GPP ranged from 0.69 to 0.79 over the Lijiang alpine meadow, which was lower than the Haibei alpine meadow (Table 7). This is the reason why the annual NEE of the Lijiang site was on the average 25% lower than the Haibei site. In general, the lower RE/GPP occurred in high-altitude and moist areas. The alpine meadow ecosystem (Lijiang and Haibei) had lower RE/GPP than most of the low-lying grasslands. Compared with semiarid grasslands (RE/GPP: approximately 1.0), the RE/GPP in the moist grasslands was much lower, e. g. the sown grassland in Netherlands (0.60) and the natural grassland in Italy (0.59) (Gilmanov et al., 2007).

## 5  Conclusions

The 4-year EC data from 2012 to 2015 were used to investigate the interannual variation in the NEE, GPP and RE. The key parameters for ecosystem photosynthesis and respiration were determined for the different seasons of each year. The vegetation growth (NDVI) controlled NEE$_{sat}$ on a monthly scale, and the interannual variation in Q$_{10}$ for the wet and dry seasons was small. The seasonal variation in CO$_2$ exchange was affected by the seasonal pattern of T$_a$ and the soil moisture in the spring. In the spring, low T$_a$ and drought events delayed the start time of CO$_2$ uptake. In the late wet season, the higher T$_a$ in 2014 and 2015 resulted in later grass senescence and CO$_2$ release. The annual NEE decreased with the length of the CO$_2$ uptake period, but its relationship with the NDVI was not significant. Over this alpine meadow, the HOS model suggests that most of the IAV in NEE, GPP and RE was attributed to seasonal variation in the climatic variables. On an annual scale, the annual RE increased linearly with MAT, while the annual GPP became saturated when MAT increased from 6.16°C to 6.32°C. Thus, the annual NEE decreased and then increased with MAT. The low RE/GPP at the study site was responsible for the lower annual NEE compared with some grassland ecosystem with larger GPP.

*Acknowledgements.* This study was supported by the National Natural Science Foundation of China (grant No.: 91537212, 41675013, 41661144018, 41461144001, 41305012) and the Third Tibetan Plateau Scientific Experiment: Observations for Boundary Layer and Troposphere (GYHY201406001). The staffs from Lijiang Meteorological Administration are appreciated for their help in the maintenance of the measurements.

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



Table 1 The average value of daily solar radiation ($S_{in}$, MJ m$^{-2}$ d$^{-1}$), the mean annual air temperature ($T_a$, $^\circ$C), the mean annual vapor pressure deficit (VPD, kPa), the mean annual soil water content (SWC, m$^3$ m$^{-3}$), the total amounts of precipitation (PPT, mm) for the whole year and the wet season, and the maximum value of NDVI for each year from 2011 to 2015

| variables | 2012 | 2013 | 2014 | 2015 |
|---|---|---|---|---|
| $S_{in}$ | 14.23 | 14.40 | 14.44 | 14.59 |
| $T_a$ | 5.93 | 5.92 | 6.32 | 6.16 |
| VPD | 0.32 | 0.30 | 0.32 | 0.30 |
| SWC | 0.232 | 0.227 | 0.232 | 0.233 |
| PPT (whole year) | 1190.4 | 1066.1 | 1204.8 | 1257.4 |
| PPT (wet season) | 1086.5 | 906.1 | 1092.6 | 1067.1 |
| $NDVI_{max}$ | 0.60 | 0.68 | 0.72 | 0.72 |





Table 2 The ecosystem photosynthesis parameters using equation (1) ($NEE_{sat}$: $\mu mol\ m^{-2}\ s^{-1}$, $\alpha$: $\mu mol\ m^{-2}\ s^{-1}$, $R^2$) and NDVI for each month during the wet seasons from 2012 to 2015. The regression was based on the average values of $NEE_{daytime}$ and PAR with PAR bins of 100 $\mu mol\ m^{-2}\ s^{-1}$. $NEE_{sat}(a)$ represents the mean value and the standard deviation, $NEE_{sat}^{(b)}$ and $NEE_{sat}^{(c)}$ represent the maximum and minimum values of NEEsat for each month.

| Month | $NEE_{sat}^{a}$ | $NEE_{sat}^{b}$ | $NEE_{sat}^{c}$ | $\alpha$ | $RE_{bulk}$ |
|---|---|---|---|---|---|
| June | -11.59±2.45 | -9.69 | -15.08 | -0.037±0.009 | 3.59±0.52 |
| July | -19.67±1.54 | -17.46 | -21 | -0.050±0.009 | 3.75±0.83 |
| August | -20.14±3.52 | -15.43 | -23.75 | -0.055±0.016 | 4.15±0.74 |
| September | -16.44±4.56 | -11.43 | -21.44 | -0.051±0.017 | 3.70±1.04 |
| October | -9.36±1.62 | -7.08 | -10.9 | -0.031±0.005 | 2.45±0.37 |

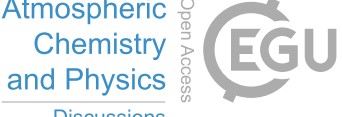



Table 3 The ecosystem respiration parameters using equation (2, 3) (a: $\mu$mol m$^{-2}$ s$^{-1}$, b, $Q_{10}$, $R^2$) for the wet and dry seasons from 2012 to 2015. The regression was based on the average values of RE and $T_{soil}$ with $T_{soil}$ bins of $1^{\circ}$C

| Season | year | a | b | $Q_{10}$ | $R^2$ |
|---|---|---|---|---|---|
| Wet season | 2012 | 0.437 | 0.125 | 3.48 | 0.98 |
| | 2013 | 0.374 | 0.124 | 3.46 | 0.94 |
| | 2014 | 0.442 | 0.126 | 3.51 | 0.98 |
| | 2015 | 0.433 | 0.123 | 3.43 | 0.98 |
| Dry season | 2012 | 0.338 | 0.081 | 2.25 | 0.78 |
| | 2013 | 0.202 | 0.096 | 2.60 | 0.74 |
| | 2014 | 0.283 | 0.115 | 3.15 | 0.99 |
| | 2015 | 0.313 | 0.104 | 2.82 | 0.70 |





Table 4 The annual total NEE, GPP and RE (g C m$^{-2}$ year$^{-1}$) for each year from 2012 to 2015

|  | 2012 | 2013 | 2014 | 2015 |
|---|---|---|---|---|
| NEE | -114.2 | -158.5 | -159.9 | -212.6 |
| GPP | 522.3 | 546.5 | 669.4 | 661.8 |
| RE | 412.1 | 393.6 | 515.2 | 456.7 |





Table 5 The percentage of the contributions of the seasonal climatic variation ($SS_s$), interannual climatic variability ($SS_i$), the ecosystem functional change ($SS_f$), and random error ($SS_e$) to the interannual variations in NEE, GPP and RE

|  | SSs | SSi | SSf | SSe |
|---|---|---|---|---|
| NEE | 37.7% | 7.7% | 10.3% | 44.3% |
| GPP | 48.6% | 9.7% | 10.7% | 31.0% |
| RE | 48.6% | 15.6% | 21.2% | 14.6% |





Table 6 The GPP$_{diff}$ for 2013-2012, 2014-2012 and 2015-2012 during the periods from March to May, June, from June to July, from August to September

| Periods | GPP$_{diff}$ | | |
| | 2013-2012 | 2014-2012 | 2015-2012 |
| --- | --- | --- | --- |
| March to May | 20.0 | 63.1 (43%) | 83.3 (60%) |
| June | 28.7 | 13.4 (9%) | 23.7 (17%) |
| July to August | -8.7 | 14.2 (10%) | -18.5 (-13%) |
| September to October | -12.0 | 55.2 (38%) | 48.2 (35%) |
| Entire year | 24.2 | 147.1 | 139.5 |





Table 7 Comparison of mean annual temperature (MAT, °C), mean annual precipitation (MAP, mm yr$^{-1}$), NEE (g C m$^{-2}$ yr$^{-1}$), GPP, RE and RE/GPP between this study and previous grassland studies

| References/Location | Ecosystem Description | Latitude | Longtitude | Altitude | MAT | MAP | NEE | GPP | RE | RE/GPP |
|---|---|---|---|---|---|---|---|---|---|---|
| This study/ Lijiang, China | Alpine meadow/shrub | 27°10'N | 100°14'E | 3560 | 6.1 | 1180 | -161 (-213 to -114) | 600 (522 to 669) | 444 (394 to 515) | 0.74 (0.69 to 0.79) |
| Yu et al., (2006)/ Damxung, China | Alpine meadow | 30°51'N | 90°05'E | 4250 | 2.1 | 520 | 28 (16 and 39) | 167 (144 and 190) | 195 (183 and 206) | 1.16 (1.08 and 1.27) |
| Kato et al. (2006)/ Haibei, China | Alpine shrub | 37°37'N | 101°18'E | 3250 | -1.0 | 566 | -121 (-193 to -79) | 634 (575 to 681) | 514 (489 to 556) | 0.81 (0.72 to 0.86) |
| Shimoda et al., (2005)/ Japan | C3/C4 grassland | 36°06'N | 140°06'E | 27 | 13.9 | 1156 | -17 (-78 to 17) | 2365 (2285 to 2426) | 2348 (2303 to 2392) | 0.99 (0.97 to 1.01) |
| Aires et al., (2008)/ Portugal | Mediterranean grassland | 38°28'N | 8°01'E | 140 | 15.5 | 669 | -71 (-190 and 49) | 893 (524 and 1261) | 822 (573 and 1071) | 0.92 (0.85 and 1.09) |
| Jensen et al., (2017)/ Denmark | Meadow | 55°55'N | 8°24'E | 0 | 8.7 | 809 | -156 (-356 to -18) | 1349 (1147 to 1570) | 1193 (1069 to 1406) | 0.88 (0.75 to 0.98) |
| Gilmanov et al., (2007)/ Europe | Multiple (19 sites) | - | - | -0.7 to 1770 | 3.9 to 14.6 | 387 to 1816 | -150 (-653 to 171) | 1261 (467 to 1874) | 1111 (493 to 1622) | 0.90 (0.59 to 1.14) |
| Xu and Baldocchi, (2004)/ USA | Mediterranean grassland | 38°24'N | 120°57'E | 129 | 16.3 | 559 | -52 (-132 and 29) | 798 (729 and 867) | 747 (735 and 758) | 0.94 (0.85 and 1.04) |
| Flanagan et al., (2002)/ Canada | Temperate grassland | 49°26'N | 112°34'E | 951 | - | 378 | -2 (-21 and 18) | 280 (272 and 287) | 278 (267 and 290) | 1.0 (0.93 and 1.07) |





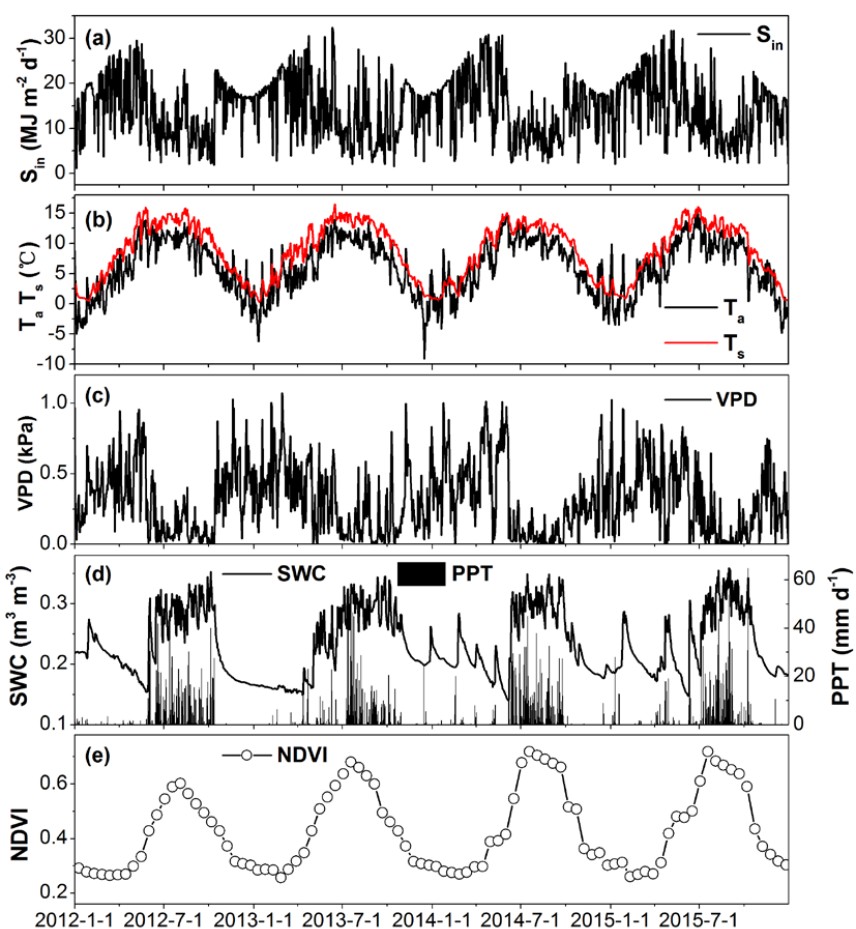

Figure 1 (a) daily sum of solar radiation ($S_{in}$), daily mean (b) air temperature ($T_a$), soil temperature ($T_s$), (c) vapor pressure deficit (VPD, 5 cm) and (d) soil water content (SWC, 5 cm), daily total precipitation (PPT), (e) 16-day average normalized difference vegetation index (NDVI) from 2012 to 2015.




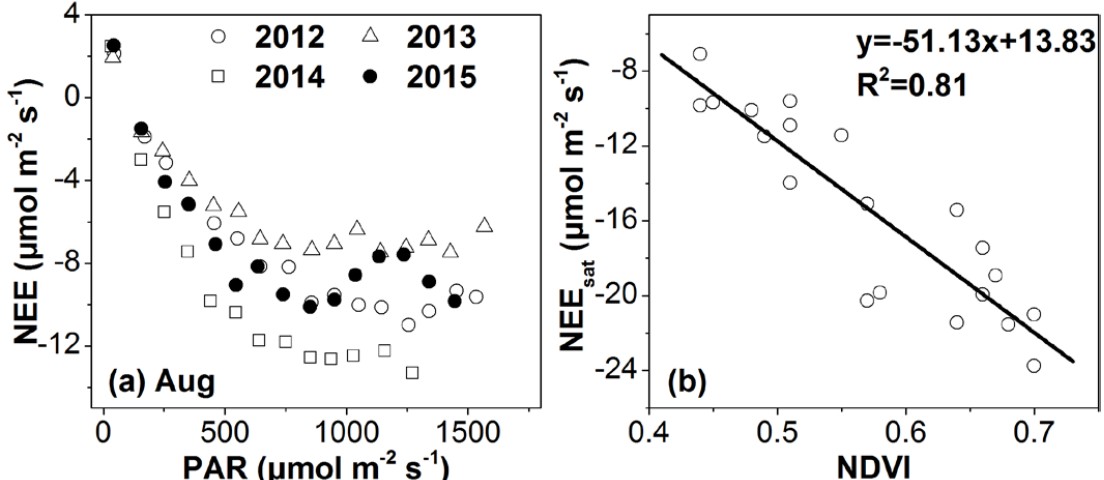

Figure 2 The relationship between daytime NEE and PAR (a) for August from 2012 to 2015. The NEE and PAR data were averaged with PAR bins of 100 $\mu mol\ m^{-2}\ s^{-1}$. (b) the relationship between $NEE_{sat}$ and NDVI on a monthly scale.





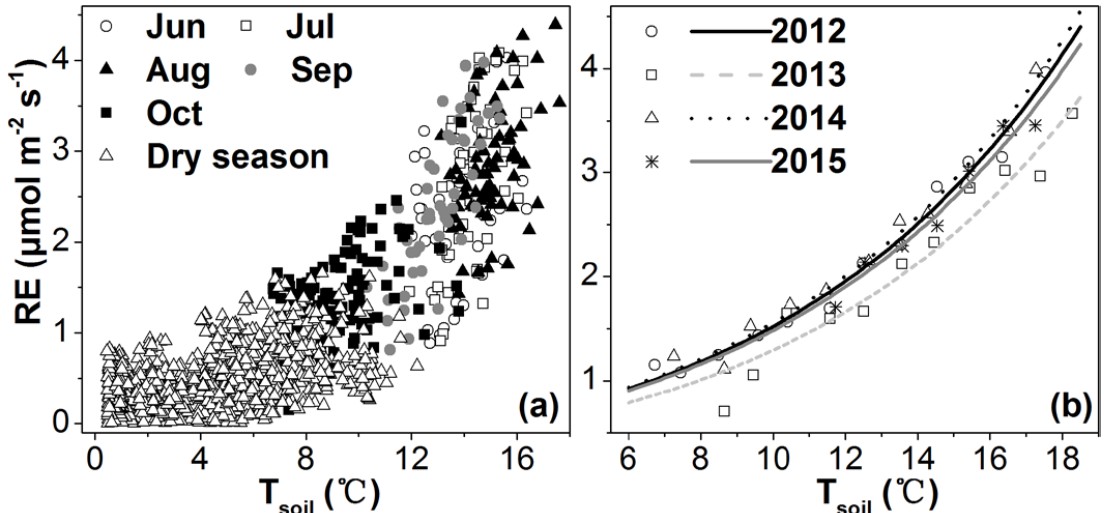

Figure 3 (a) relationship between RE and $T_{soil}$ in 2012; (b) relationship between RE and Tsoil for the wet season from 2012 to 2015; RE and $T_{soil}$ were avreaged with $T_{soil}$ bins of $1^{\circ}$C.



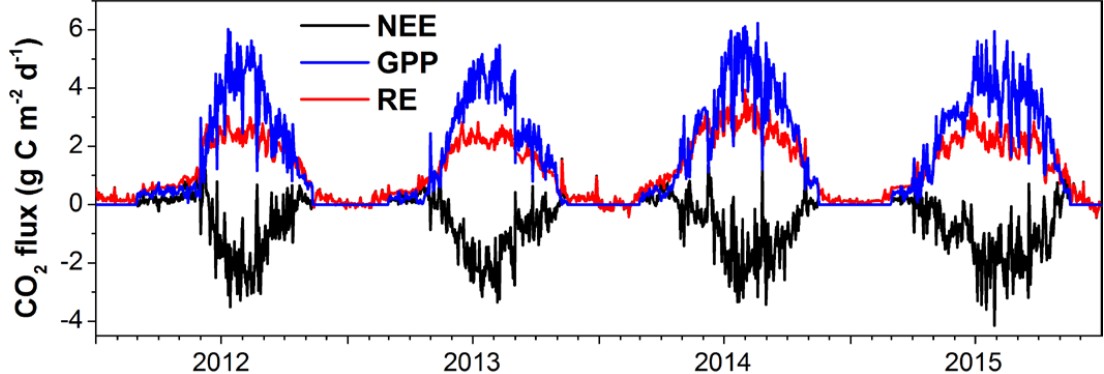

Figure 4 The daily mean NEE, GPP and RE from 2012 to 2015





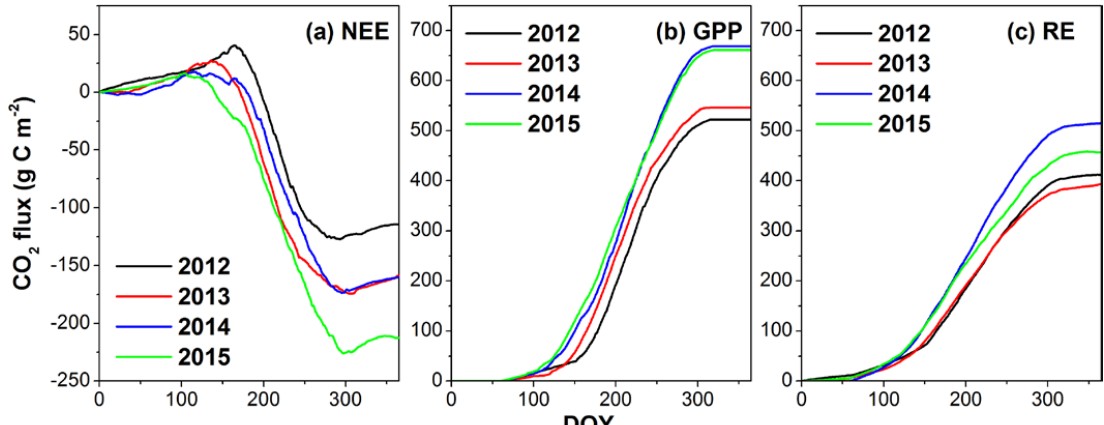

Figure 5 The cumulative NEE, GPP and RE from 2012 to 2015





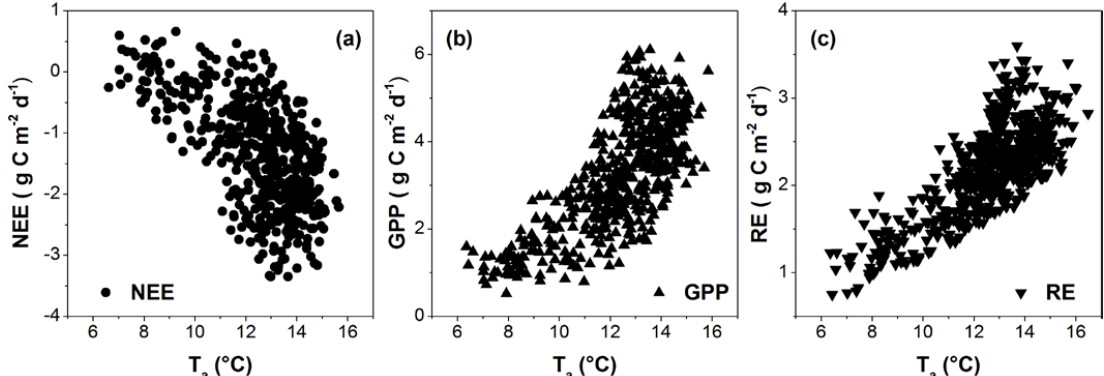

Figure 6 Relationships between (a) NEE and $T_a$, (b) GPP and $T_a$, and (c) RE and $T_a$ for the wet seasons from 2012 to 2015




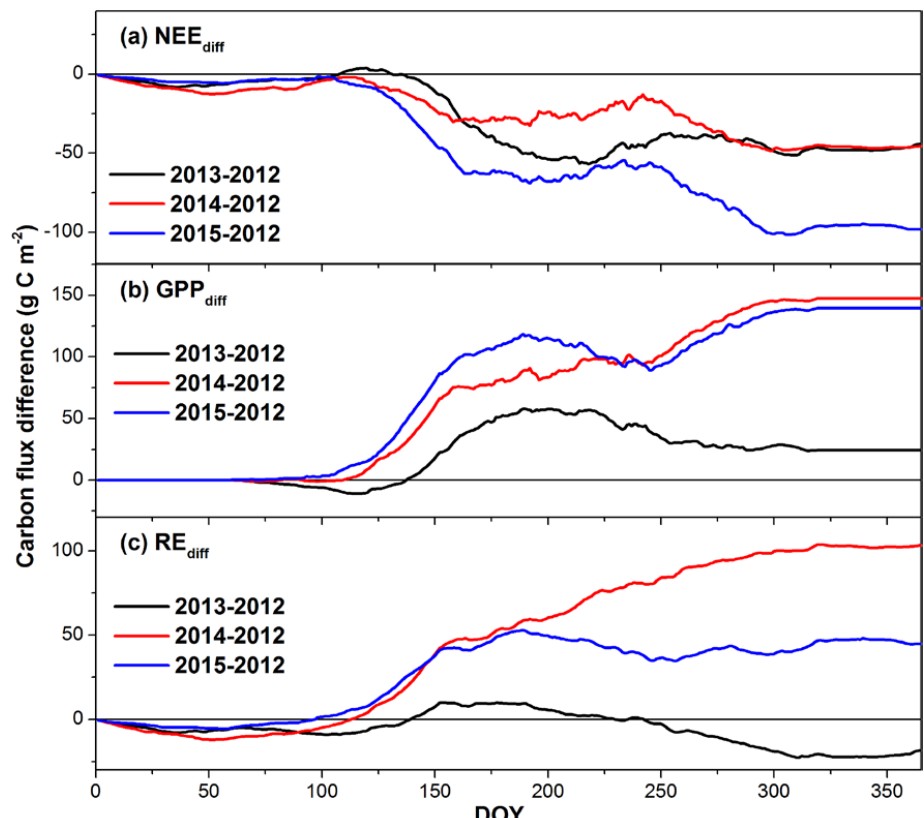

Figure 7 Seasonal variations in the differences of (a) NEE, (b) GPP and (c) RE from 2013 to 2012, from 2014 to 2012 and from 2015 to 2012





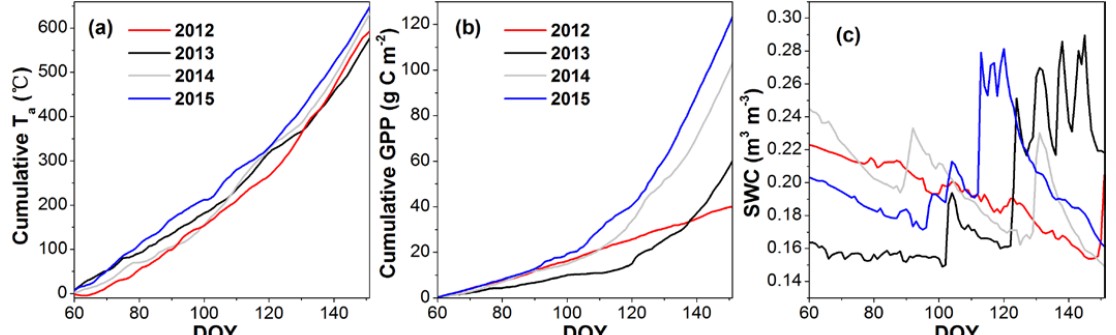

Figure 8 Cumulative (a) $T_a$ and (b) GPP, and (c) the daily mean SWC from March to May (DOY60 to 151) for 2012, 2013, 2014 and 2015





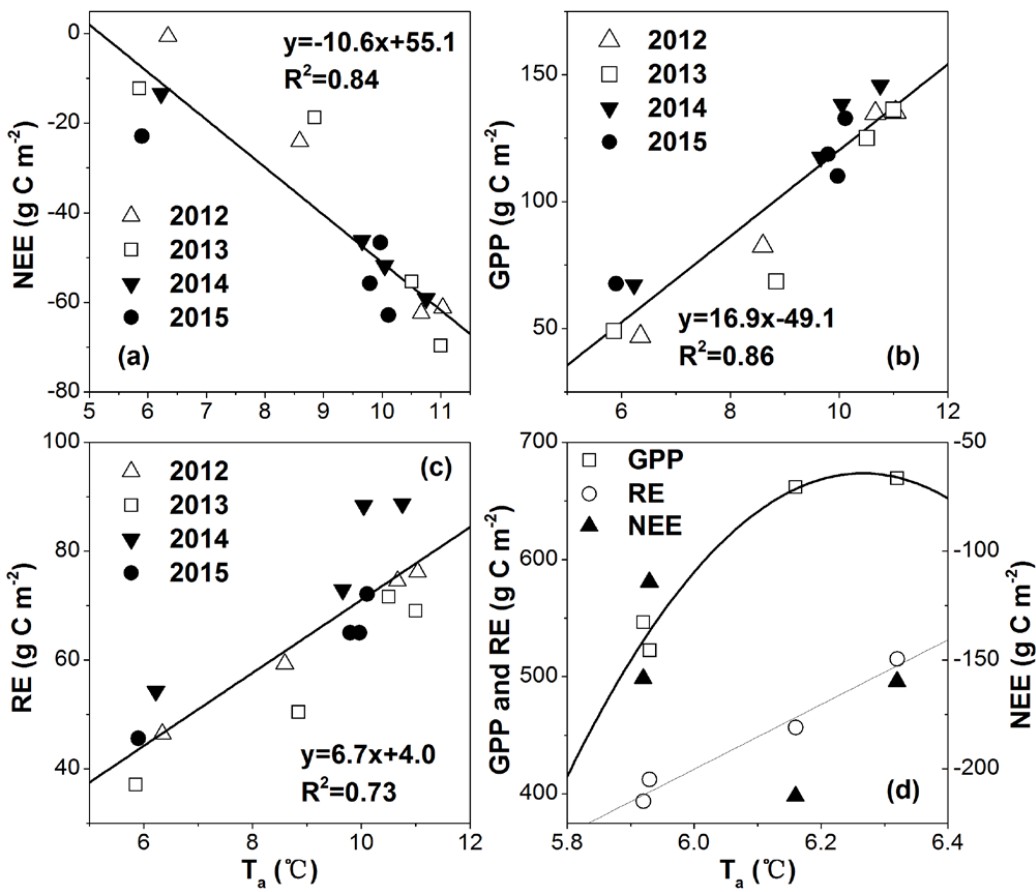

Figure 9 Relationships between (a) NEE and $T_a$; (b) GPP and $T_a$; (c) RE and Ts from July to October at a monthly scale, and (d) relationship between the annual total $CO_2$ exchange fluxes and the mean annual $T_a$, which were $GPP=-1191T_a^2+14930T_a-46102$, $R^2=0.97$, and $RE=276T_a-1235$, $R^2=0.97$