# Peer review of "Biophysical effects on the interannual variation in carbon"

_Atmospheric Chemistry and Physics, 2016_

## Referee Comment (RC1) · Anonymous Referee #1 · 12 Jan 2017

General Comments

This manuscript presents 4-year data set of $CO_2$ exchange for a high elevation grassland on the southeast margin of Tibetan Plateau. The statistic model HOS is used to partition the inter-annual variability in net ecosystem exchange between climatic variability and functional change. The annual patterns and inter-annual variability of NEE were showed in this study too. Many studies have revealed the relationship between the climate variables and the $CO_2$ exchange. This paper means to discuss the biophysical effects on inter-annual variation in $CO_2$ exchange. It is supposed to give us some new understandings. However, the authors just partition the climatic and biotic effects and give more analysis on the climatic effects. The key point should be focus on

the biophysical effects on the inter-annual variability in NEE. The authors should make more effort on revising this manuscript.

Specific Comments

1.Page 2 Introduction There are some flaws in the consistency in this section. The English writing should be improved attentively. Though the manuscript is understandable, it reads awkwardly in some sentences due to the structure or the chosen word. The authors should make more effort English writing for the entire paper.

2.Page 3 The site is in Yunnan Province, locates on the southeast of the Tibetan Plateau. The climate condition, such as annual precipitation and mean air temperature, is quite different with Tibetan Plateau. This alpine meadow has limited similarity with the grasslands on Tibetan Plateau. Thus, the site location should be described more specific in the title.

3.Page 2 line 48 The phrase (global warming) appears abruptly here. The author should explain unambiguously what they want to express.

4.Page 6 line170 There were many study on grasslands in Tibetan Plateau. The author can compare the study with other results of different alpine grasslands on Plateau. Line 174 I think the authors mean the ecosystem became a carbon sink when daily NEE was negative. The date when the ecosystem stared to absorb $CO_2$ was much earlier than the date when the negative value of daily NEE appeared. The expression in this paper should be more precise.

5.Page 7 The HOS model was interpreted in detail in Hui's paper which was published in 2003. However, I think the models and the abbreviation (SSf, SSi, SSs, SSe) should be briefly and clearly introduced in this paper. Otherwise, the readers must find out Hui's paper and figure out what the models and abbreviations mean. The variation of REdiff 2014-2012 was quite different with the other REdiff. How the authors explain this result?

6.Page 8 The authors compared NEE and the ration of RE/GPP of this alpine meadow with other grasslands. Several values were listed in this paragraph. However, more discussion is necessary. What's the purpose for this comparison? How the difference occurs?

Table 7 How many years' data were used to obtain the mean annual variables in different sites?

Actually, the discussion section is short and some paragraph is still describing the result. Please go deep into the results and present more discussion on the possible reasons of the phenomenon. The authors pay more attention on the climatic variables. However, the focus should be on the 'biophysical effects' as stated in the title.

---

## Referee Comment (RC2) · Anonymous Referee #2 · 30 Jan 2017

The authors studied the interannual variation in carbon dioxide exchanges and its controls over an alpine meadow site in the Tibetan Plateau. They found that the GPP were mainly related with Ta during the wet season, and seasonal variation of climatic factors largely affect the GPP variability. The manuscript may require substantial improvements before possible publications. Some specific comments are as follows.

1. The paper is mainly about investigating the environmental controls of the GPP. What does the "Biophysical effects" in the title mean here?

2. Although the author made large efforts in explaining the experiments and analyzing the data, they need to elaborate and highlight the key innovation of the study (e.g., new concepts, ideas, methods, or data)?

[Figure]
**Interactive comment**

3. The method used in this study is largely empirical, and most of the findings in the study are already well-known to the scientific community as the effects of temperature on GPP have been well simulated in land surface models. The author may need to consider what makes the study be different from existing studies given a special alpine meadow site.

4. The authors only used 4-year data from an individual site. Could the data be representative of the alpine meadow in the Tibetan Plateau? And if other scientists want to reproduce their results, are there any ways to obtain the data?

5. It needs more explainations on the processing of the NDVI data. It seems to me that there are no considerations for cloud-contaminated data or BRDF-affected data.

6. Although the general structure of the paper is clear, the manuscript requires large improvements on the texts. There are apparent grammar errors that need to be corrected.

---

## Author Response (AR1)

**Response to RC1**

General Comments

This manuscript presents 4-year data set of CO2 exchange for a high elevation grassland on the southeast margin of Tibetan Plateau. The statistic model HOS is used to partition the inter-annual variability in net ecosystem exchange between climatic variability and functional change. The annual patterns and inter-annual variability of NEE were showed in this study too. Many studies have revealed the relationship between the climate variables and the CO2 exchange. This paper means to discuss the biophysical effects on inter-annual variation in CO2 exchange. It is supposed to give us some new understandings. However, the authors just partition the climatic and biotic effects and give more analysis on the climatic effects. The key point should be focus on the biophysical effects on the inter-annual variability in NEE. The authors should make more effort on revising this manuscript.

**Response**: We would like to thank anonymous Referee #1 for his more detail valuable comments on this manuscript. It is very helpful to improve this paper. The manuscript is revised to focus on the biophysical effects on the inter-annual variability in NEE. Responses to all the points raised by the referee are in the following.

Specific Comments

1.Page 2 Introduction There are some flaws in the consistency in this section. The English writing should be improved attentively. Though the manuscript is understandable, it reads awkwardly in some sentences due to the structure or the chosen word. The authors should make more effort English writing for the entire paper.

**Response**: The English writing has be improved for the entire paper.

2.Page 3 The site is in Yunnan Province, locates on the southeast of the Tibetan Plateau. The climate condition, such as annual precipitation and mean air temperature, is quite different with Tibetan Plateau. This alpine meadow has limited similarity with the grasslands on Tibetan Plateau. Thus, the site location should be described more specific in the title.

**Response**: The site location is described in the abstract (L14). The title seems too long if the specific site location is given in the title.

3.Page 2 line 48 The phrase (global warming) appears abruptly here. The author should explain unambiguously what they want to express.

**Response**: The phrase (global warming) has been deleted.

4.Page 6 line170 There were many study on grasslands in Tibetan Plateau. The author can compare the study with other results of different alpine grasslands on Plateau. Line 174 I think the authors mean the ecosystem became a carbon sink when daily NEE was negative. The date when the ecosystem stared to absorb CO2 was much earlier than the date when the negative value of daily NEE appeared. The expression in this paper should be more precise.

**Response**: More results of different alpine grasslands on the Plateau have been added in the revised manuscript (section 4.3). The expression in this paper has be revised to be more precise.

5.Page 7 The HOS model was interpreted in detail in Hui's paper which was published in 2003. However, I think the models and the abbreviation (SSf, SSi, SSs, SSe) should be briefly and clearly introduced in this paper. Otherwise, the readers must find out Hui's paper and figure out what the models and abbreviations mean. The variation of REdiff 2014-2012 was quite different with the other REdiff. How the authors explain this result?

**Response**: The introductions of the models and the abbreviation have be added in the revised manuscript (section 4.3). The RE in 2014 was the largest because both the soil temperature and $Q_{10}$ in 2014 were larger than the other years.

**Response to RC2:**

The authors studied the interannual variation in carbon dioxide exchanges and its controls over an alpine meadow site in the Tibetan Plateau. They found that the GPP were mainly related with Ta during the wet season, and seasonal variation of climatic factors largely affect the GPP variability. The manuscript may require substantial improvements before possible publications. Some specific comments are as follows.

1. The paper is mainly about investigating the environmental controls of the GPP. What does the "Biophysical effects" in the title mean here?
**Response**: Biophysical effects in the title mean the controls of the biological and environmental variables.

2. Although the author made large efforts in explaining the experiments and analyzing the data, they need to elaborate and highlight the key innovation of the study (e.g., new concepts, ideas, methods, or data)?
3. The method used in this study is largely empirical, and most of the findings in the study are already well-known to the scientific community as the effects of temperature on GPP have been well simulated in land surface models. The author may need to consider what makes the study be different from existing studies given a special alpine meadow site.
**Response**: The Lijiang alpine meadow is a new site on the southeast margin of the Tibetan Plateau. The annual precipitation for this site is much larger than other sites on the Tibetan Plateau. The data tested how the $CO_2$ exchange of alpine meadow change when the annual precipitation increases from around 600 to over 1000 mm $yr^{-1}$ in China (Figure 10).

4. The authors only used 4-year data from an individual site. Could the data be representative of the alpine meadow in the Tibetan Plateau? And if other scientists want to reproduce their results, are there any ways to obtain the data?
**Response**: The Lijiang alpine meadow is located on the southeast margin of the Tibetan Plateau. However, the $CO_2$ exchange of this meadow is different from some other sites on the Tibetan Plateau, such as the Haibei alpine meadow (Kato et al. 2006) and the Damxung site (Fu et al., 2009). The inter-site comparison will help other scientists to reproduce the results.

5. It needs more explanations on the processing of the NDVI data. It seems to me that there are no considerations for cloud-contaminated data or BRDF-affected data.
**Response**: More explanations on the processing of the NDVI data have been added (L125).

6. Although the general structure of the paper is clear, the manuscript requires large improvements on the texts. There are apparent grammar errors that need to be corrected.

**Response**: The manuscript has been improved on the texts.

[revised manuscript text omitted]